# The Molecular Determination of Hybridity and Homozygosity Estimates in Breeding Populations of Lettuce (*Lactuca sativa* L.)

**DOI:** 10.3390/genes10110916

**Published:** 2019-11-09

**Authors:** Alice Patella, Fabio Palumbo, Giulio Galla, Gianni Barcaccia

**Affiliations:** Department of Agronomy, Food, Natural Resources, Animals and Environment, University of Padova, 35020 Legnaro PD, Italy; alice.patella@phd.unipd.it (A.P.); giulio.galla@unipd.it (G.G.); gianni.barcaccia@unipd.it (G.B.)

**Keywords:** pure lines, F1 hybrids, microsatellite markers, marker-assisted breeding, crop improvement, varieties

## Abstract

The development of new varieties of horticultural crops benefits from the integration of conventional and molecular marker-assisted breeding schemes in order to combine phenotyping and genotyping information. In this study, a selected panel of 16 microsatellite markers were used in different steps of a breeding programme of lettuce (*Lactuca sativa* L., 2 *n* = 18). Molecular markers were first used to genotype 71 putative parental lines and to plan 89 controlled crosses designed to maximise recombination potentials. The resulting 871 progeny plants were then molecularly screened, and their marker allele profiles were compared with the profiles expected based on the parental lines. The average cross-pollination success rate was 68 ± 33%, so 602 F1 hybrids were completely identified. Unexpected genotypes were detected in 5% of cases, consistent with this species’ spontaneous out-pollination rate. Finally, in a later step of the breeding programme, 47 different F3 progenies, selected by phenotyping for a number of morphological descriptors, were characterised in terms of their observed homozygosity and within-population genetic uniformity and stability. Ten of these populations had a median homozygosity above 90% and a median genetic similarity above 95% and are, therefore, particularly suitable for pre-commercial trials. In conclusion, this study shows the synergistic effects and advantages of conventional and molecular methods of selection applied in different steps of a breeding programme aimed at developing new varieties of lettuce.

## 1. Introduction

Lettuce (*Lactuca sativa* L.) is a self-pollinating leafy vegetable species (2 *n* = 2 *x* = 18) of the Asteraceae family. It is cultivated on a large scale throughout the world for consumption as a fresh vegetable on its own or in combination with other ready-to-eat vegetable products [1]. Its growing economic importance has led seed companies to regularly develop new varieties with ever higher agronomic traits. However, breeding programmes are highly limited by the reproductive system of this species. The flower structure of lettuce determines a reproductive strategy known as cleistogamy, in which anther dehiscence and subsequent pollination take place before flower opening, resulting in a very high rate of self-pollination, very often equal to or close to 100% [2]. According to recent estimates, out-cross rates are limited to 1–6% [3]. These reproductive barriers mean that in natural conditions the species spontaneously constitutes pure lines, characterised by phenotypic uniformity and genotypic stability, due to their very high homozygosity. In conventional breeding programmes, developing segregating and recombinant F2 populations traditionally requires crosses to be hand pollinated while self-pollination is prevented by emasculating the flowers. The most popular emasculation and hand-pollination technique is that described by Olivier [4]. Known as the “wash method”, it involves hand-spraying the inflorescence with water during pistil emergence to remove the pollen attached to the female part of the flower. The inflorescence is then left to dry for a short period, after which it is rubbed with a ripe flower of the pollinating variety [5]. A slightly different, but also widely used, technique is the “clip-and-wash method”, which involves clipping the tip of the corolla before spraying with water. This guarantees more efficient pollen removal and cross rates close to 100% from the subsequent manual pollination [2]. However, these breeding techniques are time-consuming and technically highly demanding, and are only really effective if coupled with molecular analyses aimed at screening progeny plants and assessing their hybridity.

In recent years, many seed firms have begun using molecular markers to carry out assisted selection schemes and to speed up varietal development programmes [3]. Simple Sequence Repeat (SSR) markers are, so far, the most commonly used markers for these purposes [6,7,8] as they are codominant, have high reproducibility and multi-allelism, and can be detected at any stage of plant development, without being influenced by the environment [9]. There are a considerable number of SSR markers for lettuce in the literature [10]. Truco, et al. [11] produced an integrated genome map from 7 different linkage maps, which included 130 SSR loci organised in 9 linkage groups. Rauscher and Simko [10] augmented this genomic map with 54 genomic SSR and 52 EST-SSR (Expressed Sequence Tag) loci. Finally, with the publication of the *L. sativa* genome draft [12], tens of thousands of new SSR regions have become available for testing and use.

Given the availability of markers in lettuce, Marker-Assisted Selection (MAS) has started to be adopted in plant breeding programmes for various purposes, including identification of resistance genes [13,14] or Quantitative Trait Loci (QTLs) of phytopathogens [15,16], the study of QTLs controlling complex traits [17,18], and investigation of the genetic identity and purity of either experimental or commercial lines [19]. On the other hand, very few attempts have been made to prove the efficiency of molecular markers in Marker-Assisted Breeding (MAB) activities, where the genotypic background is molecularly investigated to complement traditional phenotypic selection [20].

In this work, SSR markers were used in three different steps of a conventional breeding scheme aimed at developing new varieties characterised by distinctiveness, uniformity, and stability (Figure 1).

We first examined the genetic background of a number of superior pure lines in order to plan experimental matings to produce F1 hybrids and then derived F2 progenies manifesting morphological variability as a result of genetic segregation and recombination (Figure 1). Each offspring in the F1 generation was analysed to distinguish the individuals resulting from planned out-crosses from those resulting from accidental selfing (Figure 1). After genotyping, the S1 individuals were discarded, and the F1 individuals were self-pollinated. In the F3 generations (Figure 1); experimental populations, previously selected according to their morphological traits, were also characterised by molecular markers due to the need to assess their stability and uniformity in order to run pre-commercial trials.

## 2. Materials and Methods

### 2.1. Plant Materials and Breeding Techniques

Plant materials were developed and provided by Blumen Group SpA, Italy and belonged to five different lettuce cultivar types (Appendix A). Seventy-one parental lines (germplasm composed of experimental, pre-commercial and commercial lines) were involved in 89 combinations of crosses, in which each progeny consisted of 6–12 individuals (871 progeny samples). Parental lines were grown in the spring of 2015, and the 89 programmed crosses were carried out in the summer using the clip-and-wash method [2]. This involved making an incision in the calyx and corolla and washing the anthers in the early morning before the pollen grains could settle naturally on the outermost stigmatic surface of pistils. The plants were then manually pollinated by rubbing anthers of the pollen donor on the stigma of the seed parent. For each planned cross, a bulk of 4/5 flowers from a pollinator parent was used to pollinate as many flowers of a seed plant. Seeds were collected from the seed plant and sown in early autumn for genotyping selection and agronomic evaluation (spring 2016).

Finally, to assess the uniformity of the 47 experimental lines, previously chosen for morpho-phenological traits and pathogen resistances, 940 samples belonging to the 47 F3 populations (labelled 1 to 47) were collected in the spring of 2018. Each experimental line comprised 20 individuals.

### 2.2. DNA Isolation

A total of 100 mg of fresh leaves was collected from each of the 1882 lettuce samples (71 parents, 871 progeny and 940 F3) and ground to a fine powder using Tissue Lyser II (Qiagen, Valencia, CA, USA). Genomic DNA (gDNA) was extracted with the Dneasy^®^ 96 Plant Kit (Qiagen), according to the manufacturer’s protocols. After extraction, the integrity of the gDNA was assessed by electrophoresis on 1% (*w/v*) agarose gel stained with SYBR Safe^®^ 1 × DNA Gel Stain (Life Technologies, Carlsbad, CA, USA) in Tris-Acetate-EDTA (TAE) running buffer. Both the yield and purity of the extracted gDNA samples were evaluated using a NanoDrop 2000c UV-Vis Spectrophotometer (Thermo Scientific, Carlsbad, CA, USA). Following DNA quantification, all DNA samples were diluted to a final concentration of 20 ng/μL. 

### 2.3. Primer Design and Testing of SSR Marker Amplification

Sixteen SSR marker loci were selected from those available in the scientific literature [10,21], according to (i) chromosomal location; (ii) polymorphism rate, expressed as PIC (Polymorphism Information Content); (iii) allele size range; (iv) annealing temperature of the locus-specific primers. Amplifications were performed according to the method previously described by Schuelke [22], with some minor modifications. Briefly, three primers were used to amplify each microsatellite locus: a pair of locus-specific primers, one with an oligonucleotide tail at the 5′ end (M13, PAN-1, PAN-2 or PAN-3, Appendix A), and a third universal primer complementary to the tail and labelled with a fluorescent dye (6-FAM, VIC, NED, or PET). Primer pair efficiency was tested in silico using the PRaTo [23] web-tool and were organised in three multiplex reactions, as shown in Table 1.

The 16 primer pairs were first tested individually (singleplex reactions) using three randomly chosen lettuce gDNA to evaluate primer efficiency and to check the correspondence between expected and actual size of the bands; they were then evaluated in multiplex PCRs to assess possible primer interactions.

All amplification reactions (both singleplex and multiplex) were performed in a 10 µL reaction volume containing 1× Platinum^®^ Multiplex PCR Master Mix (Thermo Scientific), 10% GC Enhancer (Thermo Scientific), 0.25 µM of non-tailed primer, 0.75 µM of tailed primer, 0.50 µM of fluorophore-labelled primer (universal primer) and 20 ng of genomic DNA. Thermal cycling conditions were as follows for multiplex 1 and 2:94 °C for 5 minutes followed by 6 cycles of 94 °C for 30 seconds, 61 °C for 30 seconds, 72 °C for 45 seconds, with a 1 °C annealing temperature stepdown per cycle (from 61 °C to 56 °C). The annealing temperature for the following 35 cycles was set at 56 °C, with denaturation and extension phases as above and a final extension at 60 °C for 30 minutes. The multiplex 3 thermal cycling conditions were instead: 94 °C for 5 minutes followed by 6 cycles of 94 °C for 30 seconds, 56 °C for 30 seconds, 72 °C for 45 seconds, with a 1 °C annealing temperature stepdown per cycle (from 56 °C to 51 °C). The annealing temperature for the following 35 cycles was set at 51 °C with denaturation and extension phases as above and a final extension at 60 °C for 30 minutes. All amplifications were performed in a GeneAmp^®^ PCR 9700 thermal cycler (Applied Biosystems, Carlsbad, CA, USA). PCR products were first checked on gel electrophoresis (2% Ultrapure™ Agarose in TAE 1×, SYBR Safe^®^ 1×, Life Technologies) then run on capillary electrophoresis with ABI 3730 DNA Analyzer (Applied Biosystem), using LIZ500 as the molecular weight standard. The size of each peak was determined with the Peak Scanner 1.0 software (Applied Biosystems).

### 2.4. Genotyping and Data Analysis

The 71 potential parents were genotyped at 16 SSR loci and statistical analyses were performed using NTSYS (Numerical Taxonomy and Multivariate Analysis System) version 2.2 (Exeter Software) [24]. Rohlf’s (or the simple matching) coefficient was used to calculate pairwise genetic similarity (GS) in all possible comparisons and to construct a genetic similarity matrix, according to the formula:GS_ij_ = m/(m + n)(1)
where “i” and “j” are two different individuals, while “m” and “n” represent the number of matching and non-matching attributes, respectively. An unweighted pair group method with an arithmetic mean (UPGMA) dendrogram and a Principal Coordinates Analysis (PCoA) of parental lines were carried out using the Jaccard coefficient in the PAST software v. 3.14 with 10,000 bootstrap repetitions [25]. The genetic structure of the lines was modelled using a Bayesian clustering algorithm implemented in STRUCTURE v. 2.2 [26]. Since no *a priori* knowledge of the origin of the populations under study was available, the admixture model and then the correlated allele frequencies model were used. Ten replicate simulations were conducted for each value of K, with the number of founding groups ranging from 1 to 8, using a burn-in of 200,000 and 1,000,000 iterations. The most likely K Estimates were determined using the method described by Evanno et al. [26]. Estimates of membership were plotted as a histogram in an Excel spreadsheet. Finally, observed homozygosity (Ho) was determined with the POPGENE software [27]. 

The 89 subsequent crosses were planned according to the following criteria: (i) high genetic dissimilarity values among parents within the same lettuce cultivar type and between them, (ii) availability of informative loci able to distinguish between the resulting offspring and individuals resulting from accidental self-pollitated. Only homozygous loci for different alleles were considered informative, whereas heterozygous loci were taken into account only if the origin of the parental alleles could be clearly discerned in the progenies. The resulting offspring (871 samples) were then screened, with the analysis restricted to those SSR loci which had previously proven to be informative for hybrid detection. This made it possible to determine whether individuals belonging to a given F1 population originated from cross-pollination or self-pollination. The successful crosses (S.C.) rate of 89 was calculated as follows: S.C. = (No F1 × 100)/(No Tot − No U.G.)(2)
where “No F1” is the number of hybrid individuals, “No Tot” is the number of all individuals in a progeny population (No tot = No F1 + No U.G. + No SP) and “No U.G.” is the number of unexpected genotypes deriving from unplanned crosses.

Finally, 940 samples from the 47 F3 populations were genotyped using the previously-described panel of SSR markers. The POPGENE software [27] was used to compute the mean values of observed homozygosity for each population (3), where n is the total number of samples). In addition, the median of genetic similarity between the 47 lines was calculated using Rohlf’s coefficient, which was designed for codominant molecular markers [28,29]. Comparison of genetic similarity among ten selected populations was instead calculated using the Jaccard coefficient, in accordance with the literature [30]. Genetic similarity matrices were generated in the NTSYS software [24].
(3)$$\bar{H}o=\sum_{n}\;\mathrm{Ho}/n$$

## 3. Results

### 3.1. Parental Lines

The 16 SSR markers, organised in three multi-locus PCRs, were used firstly to amplify and score the 71 parental lines. Fourteen of the 16 SSR markers proved to be polymorphic among plant accessions. The similarity matrix constructed using Rohlf’s coefficient revealed genetic similarity values ranging from 53% to 100% (Appendix A). The resulting unweighted pair group method with an arithmetic mean (UPGMA) dendrogram showed the samples clustering into two main sub-groups. Eighteen parental lines were not fully distinguishable, while the remaining 53 had unique genotypic profiles. The first principal coordinate from the PCoA accounted for 22% of the total variation and divided the samples into two groups, analogous to the clustering in the tree. The second principal coordinate accounted for 12% of the total variation. These results were confirmed by investigation of the genetic structure of the 71 parental lettuce lines based on allele frequencies; the best estimate of population size was *K* = 2 (Appendix A), such that the samples were grouped into two genetically distinct clusters (Figure 2). The lettuce cultivar types were reported in Appendix A, but they did not correspond to different clusters in the UPGMA tree.

The mean observed homozygosity was 82%, with a minimum value of 69% and a maximum of 100%. It is worth noting that 19 of the 71 parental lines (27%) had observed homozygosity values greater than 90%, while 30 of the 71 (42%) had a medium-high observed homozygosity (Ho) between 81% and 90%. Fourteen of the 71 parental lines (20%) had observed homozygosity ranging from 71% to 80%, and only 8 individuals had values lower than 70% (Figure 3a and Appendix A).

### 3.2. Determination of Hybridity

Using a combination of genotypic and phenotypic data, 89 cross combinations were planned (Appendix A). Before proceeding, we also checked the availability of informative loci able to distinguish between offspring resulting from out-cross and those obtained by accidental self-pollination. Screening identified 1 discriminant locus in 16% of cases, 2 informative loci in 36% of cases, and 3 to 7 informative molecular markers in 48% of the crosses (Figure 3b). The three most informative loci were Lsat3, Lsat7, and Lsat6, while Lsat4 and Lsat13 were monomorphic in almost all parental groups. It is worth noting that the Lsat8 marker was in a heterozygous state in all but four parental genotypes (7, 45, 58, and 60).

We were able to take advantage of these informative loci to calculate the success rate of each cross. In 30% of manual pollinations (27 out of 89), a success rate of 100% was achieved (i.e., all the offspring were hybrids), whereas in 18% of crosses (16 out of 89), the S.C. ranged from 71% to 90%. A hybridity rate fluctuating between 51% and 70% was reported in 15% of cases (13 out of 89), while 26% of the crosses produced fewer than 50% hybrids each. Finally, in only 7% of crosses (6 out of 89) were all the offspring the result of self-pollination (Figure 3c and Appendix A). Overall, the mean hybridization rate (the average number of hybrids per crosses) was 68 ± 33%, and out of a total of 871 individuals, 602 (69%) were hybrids, and 556 were derived from programmed crosses. The remaining 46 individuals (5% of the total) had a unexpected genotypes (U.G.) compared with their putative parents (Appendix A).

### 3.3. Lettuce Breeding Populations

The 47 F3 experimental lines were genotyped using the same set of 16 SSR loci as for the previous analyses. The homozygosity estimates of all samples (940) ranged from 67% to 93% (Figure 4a). Ten experimental populations had a median observed homozygosity ≥90%. Outliers—with homozygosity values consistently deviating from the median—were present in only three experimental populations (11, 14, and 32).

The median genetic similarity observed within each line was always greater than 90% (Figure 4b), and 37 experimental populations had a median genetic similarity ≥95%. Outliers were present in 21 of the 47 lines (Figure 4b).

After assembling the data, we found 10 breeding populations, belonging to butterhead type (Appendix A), to have Ho values ≥ 90%, and a median genetic similarity ≥ 95%; the box-plots of these populations are labelled in red in Figure 4a,b. Finally, in the genetic similarity matrix calculated from all pairwise comparisons of these ten populations, the Jaccard’s index ranged from 44% ± 3% (between populations 3 and 18) to 96 ± 5% (between populations 45 and 47, Appendix A). Moreover, the populations called 45 and 47 were constituted starting from the same parents (2 × 6, Appendix A). 

## 4. Discussion

The last decade has seen major advances in the acquisition of knowledge concerning the genetics of lettuce and, in particular, the development of molecular markers [1,11,21]. This has facilitated marker-assisted selection programmes, especially those aimed at countering the onset of new diseases. For example, several studies have dealt with identifying the QTLs associated with biotic and abiotic stress resistance [17,31,32]. Molecular markers have also been extensively used to assess genetic variation and relationships in lettuce germplasm [19] and to identify possible duplicate varieties [33]. However, although the benefits derived from exploitation of these molecular tools have also been discussed in marker-assisted breeding programmes [34] and demonstrated in several species [7,35], there are only a few studies on this type in lettuce [20]. The aim of our work, therefore, was to integrate conventional and biotechnological methods in three different steps of a breeding programme to show that this strategy is also effective in *L. sativa* (Figure 1). This is of pivotal importance if we consider the economic impact of lettuce (the world production of lettuce and chicory in 2017 was 26.8 million tons [36]) and the need to regularly develop new varieties.

Commercial lettuce varieties are usually characterised by pure lines due to the autogamous nature of this species. In order to introduce variability, manual pollination is usually carried out to cross genetically stable parent lines with agronomic traits of interest. Progeny selection is a crucial step, but despite the efficiency of some emasculation and hand pollination methods developed over the years [2], a major problem—distinguishing unequivocally and rapidly F1 individuals from self-pollinated progeny—still remains. The use of molecular assays to quickly and accurately screen progeny makes it possible to overcome most of the conventional breeding limits in this species.

In this context, our SSR-based analysis has (i) facilitated selection of the best parents to cross in order to maximise the variability of the progeny both within the same cultivar type and among them, (ii) allowed accurate evaluation of the resulting offspring, and (iii) sped up the screening of experimental F3 lines for their stability and uniformity.

The first part of our work focused on pre-screening 71 parental lines for crossing with the aim of maximising the gains obtained from each out-pollination within cultivar type and, in some cases, among them. As expected, the similarity matrix and the unweighted pair group method with an arithmetic mean (UPGMA) dendrogram showed varying levels of similarity among the different parental genotypes. Parental germplasm appeared to divide into two different groups, as revealed by the Principal Coordinates Analysis (PCoA) results and particularly by the genetic structure analysis. However, samples did not separate in UPGMA tree and PCoA according to the cultivar type, but we may assume that increasing the number of markers it could be possible to clarify this clustering. Although 53 parental lines were found to be fully distinguishable, with similarity values ranging from 53% to 98% and characterised by a unique genotypic profile, it was impossible to identify unequivocally the remaining 18. This is not surprising if we consider that some of the parental lines were closely related. We may speculate that increasing the number of SSRs would allow us to address these remaining issues. Given the aim of this study, these data were useful to avoid crosses between parents with 100% similarity. To introduce variability according to the phenotype and the lowest similarity values, we carried out 89 crossing combinations. Another aspect that needs to be considered when planning crosses is the stability of the parental line in terms of homozygosity. In our study, the median observed homozygosity of the parental lines was lower than expected (82%), especially in light of the strictly autogamous reproductive system of lettuce [37]. Overall, the fact that only one individual in four had homozygosity values greater than 90% showed that some of these lines were not entirely stable. However, it must be borne in mind that, although the observed homozygosity was not optimal, some of these lines, experimental lines, were chosen to produce F1 partly because they displayed resistance to multiple pathogens and had interesting phenotypic traits.

Before proceeding with hand pollination, in order to distinguish between F1 individuals resulting from cross-pollination and those resulting from self-pollination, we first examined the informative loci among the parental lines used in the crosses. Only homozygous loci for different alleles in parental lines were considered informative. Our analysis showed at least 2 informative loci in 84% of the programmed crosses. It is worth pointing out that restricting the analysis to the informative loci brought us considerable savings in terms of time and costs.

Overall, the molecular determination of hybridity was successful: F1 individuals represented at least 51% of the offspring in 67% of the manual crosses, and 100% of the offspring in 30% of the crosses (100% success rate), in agreement with the estimates originally reported by the developer of the pollination technique [2]. Unexpected genotypes (U.G.) were identified in 5% of the individual progeny. In these cases, the progeny genotypes appeared to diverge from what would be expected given the parents. This percentage is consistent with the spontaneous or undesired occurrences of cross-pollination (1–6%) reported in the literature for this species [3], mainly due to pollinator insects. However, we cannot exclude human error during manual pollination or seed collection.

Finally, at an advanced step of the breeding programme, we genetically assessed 47 different experimental F3 populations (940 samples), previously selected for their morpho-phenotypic traits and resistances of interest (Figure 1). Interestingly, the findings in terms of both homozygosity and intra-line similarity were very good. This would suggest that in strictly autogamous species, such as lettuce, three cycles of self-pollination may already be sufficient to reach desired outcomes in terms of genetic uniformity and homozygosity. This also confirms that the use of molecular markers could speed up the process by making it possible to select the best individuals on the basis of their genotype, thereby reducing the number of generations needed to develop new varieties. The ten experimental populations with the highest homozygosity estimates (≥90%) and the highest intra-genetic similarity values (≥95%) were considered suitable for pre-commercial trials (red box plot, Figure 4). However, a pairwise comparison of two of them (identified as 45 and 47) showed them to be genetically too similar (96% ± 5% genetic similarity, Appendix A), in agreement with phenotypic data and their common origin (Appendix A), to be registered and marketed as distinct varieties. According to the most recent guidelines concerning the protection of new plant varieties, the similarity threshold to define two lettuce varieties as distinct is set at 96% [30]. The next step will be to integrate molecular data and morphological observations in order to the select the best genotypes (positive selection) for evaluation as pre-commercial varieties. In particular, the eligible genotypes will be self-pollinated to multiply the seed so that their agronomic performance can be compared in different locations and periods of the year, and with the best commercial varieties already on the market.

For the remaining experimental populations (white box blot, Figure 4), an attempt could be made to increase their genetic uniformity through negative selection to remove the most genetically divergent individuals (i.e., outlier samples). Moreover, if necessary, the remaining genotypes can undergo a further selfing cycle aimed at reaching optimum values of homozygosity.

## 5. Conclusions

The results of this study demonstrate the advantages of mutual integration of traditional and biotechnological methods and show the added value that molecular markers can give to breeding programmes. We used microsatellite markers in three different steps of a conventional lettuce breeding program (see Figure 1) and demonstrated, firstly, the efficiency of SSR markers not only in selecting the best parental plants for crossing based on their observed homozygosity and dissimilarity values, but also in screening the resulting F1 progeny to distinguish between the offspring resulting from cross-pollination and those resulting from self-pollination. Furthermore, using the same SSR panel, we were able to act downstream of the breeding scheme to assess the uniformity of some pre-commercial cultivars. Our molecular assay could therefore also be used by seed firms to assess newly developed varieties for distinctiveness, uniformity and stability (DUS test), three major requirements for registering plant materials [6]. Finally, molecular characterisation of a new variety could also be used to register it in national or international varietal catalogues. In fact, the genotype or molecular profile of a registered variety can be crucial in solving cases of fraudulent practices, and in curbing plagiarism and unfair free-riding on the original plant breeder’s time and investment [30].

## Figures and Tables

**Figure 1 genes-10-00916-f001:**
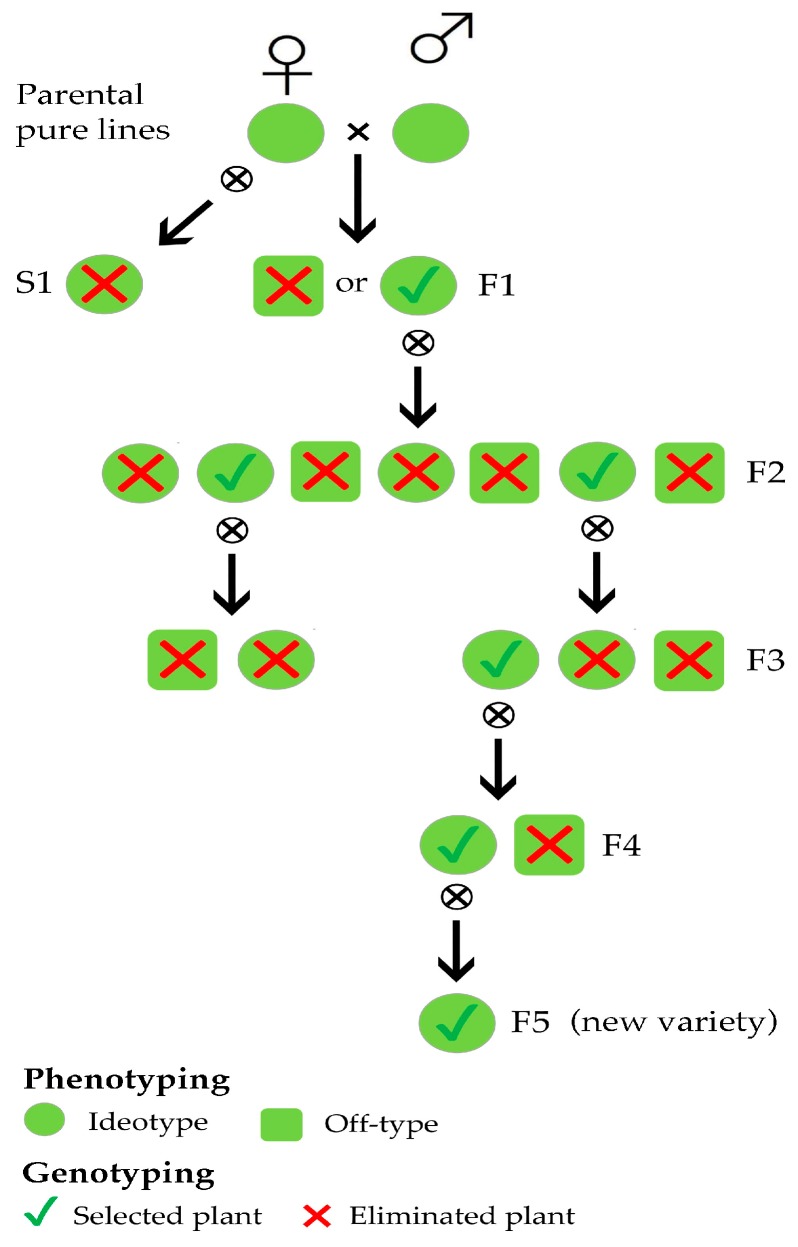
Simplified overview of a lettuce breeding scheme in which selection is based on both plant phenotyping and genotyping.

**Figure 2 genes-10-00916-f002:**
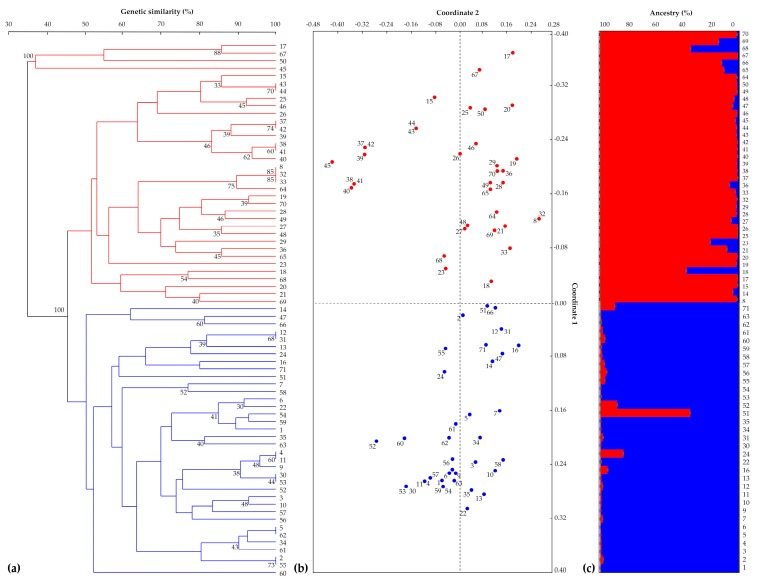
(**a**) Genetic similarity-based unweighted pair group method with an arithmetic mean (UPGMA) dendrogram of 71 parental lines calculated using the Jaccard coefficient. Bootstrap estimates ≥30% are reported next to the nodes (red and blue branches indicate the two clusters identified). (**b**) Principal coordinate analysis (PCoA). The 71 samples are shown in red or blue according to the clustering shown in the UPGMA tree. (**c**) The population genetic structure of the 71 lines as estimated by STRUCTURE. Each sample is represented by a vertical histogram partitioned into *K* = 2 coloured segments (red or blue, in accordance with (**a**,**b**)) representing the estimated membership. The proportion of subgroup membership (%) is reported on the ordinate axis, and the identification number of each accession is reported below each histogram.

**Figure 3 genes-10-00916-f003:**
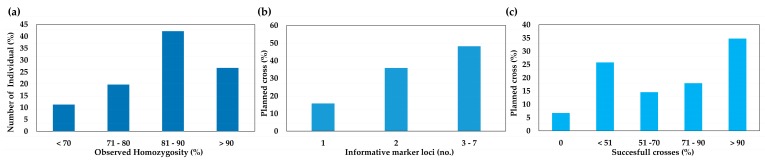
(**a**) Observed homozygosity of 71 lettuce parental individuals belonging to as many pure lines. (**b**) Histogram of discriminating loci in 89 cross combinations (in percentages). (**c**) Histogram of the percentages of pollination success in 89 programmed lettuce crosses.

**Figure 4 genes-10-00916-f004:**
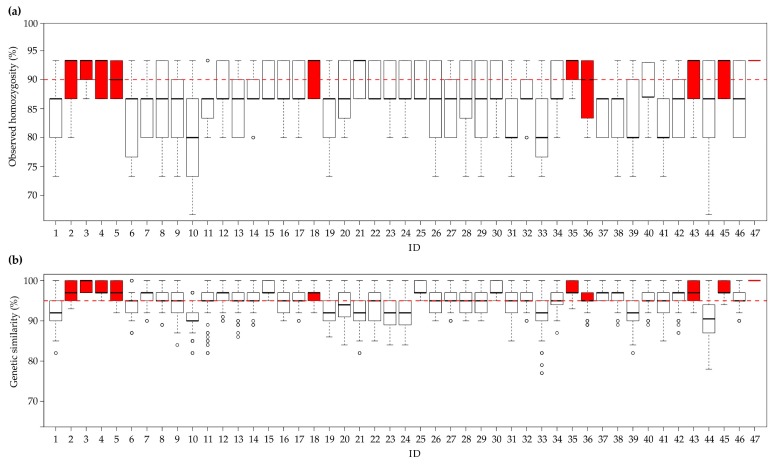
Statistics relating to the observed homozygosity and genetic similarity among lines. (**a**) Box-plot of the median observed homozygosity (in percentages) in each of the 47 populations. The red dotted line represents the homozygosity threshold set at 90%. (**b**) Box plot of the median genetic similarity in each experimental population (in percentages). The red dotted line represents the genetic similarity threshold set at 95%. The red box-plots represent the ten best experimental populations (observed homozygosity ≥90% and genetic similarity values ≥95%). The second and third quartiles are marked inside the square and are divided by a bold bar (median). Dots show outlier samples.

**Table 1 genes-10-00916-t001:** Microsatellite loci information. For each primer pair, the original simple sequence repeat (SSR) name, ID used in this work, linkage group [10,21], SSR motif, primer sequences (PAN1, PAN2, PAN3, or M13 tails at the 5’ end are indicated in square brackets; for further details see Appendix A), dye and the multiplex to which the SSR marker locus belongs is shown.

Marker Name	ID	LG	Motif	Primer Sequence		Dye	Multiplex
LSSA27-2 [10]	Lsat1	1	(AC)_7_	For	[M13]CACACTACCACCCAACACG	6-FAM	1
				Rev	ACCCTCTTCGCTTCTTCTT		
SML-045 [21]	Lsat2	2	(AAG)_9/12_	For	ACAAAACCGTTTCACCCAAA	6-FAM	1
				Rev	[M13]AGCCCTGTCCTCTTCAGGAT		
LSSB54 [10]	Lsat3	8	(GT)_10_	For	[PAN1]CTTGAGAGTGCTTGGAGAGGAT	VIC	1
				Rev	CACATACAACAAGACAAGTCCCA		
LSSA05 [10]	Lsat4	8	(TC)_18_	For	AGAACAACGGTAGCTTGTTAAATTG	VIC	1
				Rev	[PAN1]ATCGTCGGTTAATCTTCGTCG		
LSSA04 [10]	Lsat5	4	(TC)_14_	For	[PAN2]AAGGAAAGGAAGGGTTGACTTGT	NED	1
				Rev	TTGGTGAAGAAAAGAGAGAGTTT		
LSSA11 [10]	Lsat6	3	(CT)_20_	For	[PAN2]ACTCCCACTATCCTCTTTGCAT	NED	1
				Rev	GCCCACATTCTTAATCTTGTCC		
LSSA14 [10]	Lsat7	9	(AG)_18_	For	[PAN3]TGATGACTCCAGTCTTAGATACCA	PET	1
				Rev	AGTCCCCGACTATCAGTCTCA		
LSSB09 [10]	Lsat8	2	(TG)_8_	For	AGAATGAGAAGGATGAAATGGCTG	6-FAM	2
				Rev	[M13]AAACACCTTTAGCATCAAAATACCC		
SML-029 [21]	Lsat9	9	(GAG) _7/8_	For	[M13]AGCCCAGAAGAGCGTGATTA	6-FAM	2
				Rev	TGCAGGGCTCCTTGATCTAC		
LSSB17-1 [10]	Lsat10	7	(GT)_11_	For	ACTAGGGCTCTAATACAACTTGT	VIC	2
				Rev	[PAN1]TTGGCTTACAGTTATGGATTAAATG		
LSSA17 [21]	Lsat11	3	(AG)_21_	For	[PAN1]AATGTGCGTGAGAGTTTCCTTT	VIC	2
				Rev	CAAGAAGGCAGTGATGAAGTTG		
LSSA12 [10]	Lsat12	5	(GT)_11_	For	[PAN2]ACAAGGCCCAATCCTTTTCT	NED	2
				Rev	TCGAAAATTTGGAGAGAGTTTCTT		
LSSA15 [10]	Lsat13	1	(AC)_11_	For	GCCCAACCCAAGAAGAGGAG	PET	2
				Rev	[PAN3]TGGAGAGGAGTGGAGAGTGTT		
LSSA28-1 [10]	Lsat14	4	(GA)_28_	For	TTCATCTCTCTCCTCCTTCAGC	6-FAM	3
				Rev	[M13]ATCCCCATTGTCCTCCC		
LSSA21-1 [10]	Lsat15	8	(TC)_19_	For	[PAN2]TTGTACCCAGTTGTCCAAACAG	NED	3
				Rev	CAGATTGTTGCAGATTTCTTCG		
LSSB68 [10]	Lsat16	na	(CT)_20_	For	GTCTGTGTGGTTTTGGT	PET	3
				Rev	[PAN3]TGTGGTGGAGTGTGATTT		

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
