# Peer review of "The Molecular Determination of Hybridity and Homozygosity Estimates in Breeding Populations of Lettuce (Lactuca sativa L.)"

_genes, 2019, doi:10.3390/genes10110916_

Round 1

Reviewer 1 Report

The manuscript is well written and explained about the role of molecular markers to determine the distance of breeding lines of Lettuce. However, most citations were older than 10 years. I recommend authors revise their literature and update with most recent information. 

Reviewer 2 Report

The authors investigated the use of 16 SSR markers for determining lettuce hybridity and estimates of homozygosity in different generations. Generally, the manuscript is interesting, though the results would be more accurate if a larger number of molecular markers were used (only 14 of the used markers were actually polymorphic). I have some major and minor comments indicated below. The relevant line number is indicated by capital letter L.

Major:

I miss information about tested cultivars. They are only coded as numbers, but that information is useless to breeders unless actual names are provided. Cultivated lettuce is classified into several horticultural types (e.g. Romaine, iceberg, Batavia, butterhead, leaf, etc.). No information was provided what type these tested lettuces were. It is crucial, for example, when evaluating results of sub-populations and clustering. 6-12 individuals per cultivar were genotyped. Were the same individuals used for making crosses? Were all plants from the cultivar used to make crosses, or just a single one? This is very important to know, as different plants within a cultivar show a range of homozygosity, leading to different results. Homozygosity of ‘pure lines’ is very low. They are supposed to have homozygosity close to 100%, but observed values were as low as 69% with mean of 82%. Any hypothesis why it is so? I wonder if lines with homozygosity of 69% actually can be called cultivars and used in commercial production. This is even more striking when comparing to homozygosity of F3 lines that was in a quite similar range (67% - 93%). L253 (11, 13, and 33). Is this correct? I can’t see it in Fig. 5a. No information is provided on pedigree of any of the tested F3 lines. What were the parents? Did line 45 and 47 had the same/similar parents? This is not true that those lines were derived from only two ancestors. There are no data to support this hypothesis. Having two sub-populations doesn't mean that there were only two ancestors. Are those sub-populations in line with observed horticultural types or known pedigrees? L332-334, Fig 5 (and elsewhere in text). The authors need to explain in more detail how homozygosity was calculated for F3 lines. From what I understand all 16 SSRs were taken and used for genotyping 20 F3 plants per line. But it was shown before that parents of those lines differ in only a very few markers, e.g. 16% of them in only a single marker. If a F3 line with parents differing in only 1 SSR were tested, homozygosity values would be over 94% even if all 20 plants were still heterozygous in the only SSR marker that detects different alleles in the two respective parents. But this can’t be used as an argument for justification of reducing the number of generations to develop new varieties (L335). It is wrong. Authors mentioned in several places that parents with highest genotypic differences may selected for crossing. If this rule is applied across all lettuce cultivars, one would always make crosses only between cultivars from different horticultural types, as they show the biggest genotypic difference. But that would lead to very difficult breeding program when true to the type material (e.g. iceberg) needs to be developed. Then, a genetically more similar parents need to be usually selected.

Minor:

L34: delete “and reputation”. L52: If phenotypes substantially differ, F1 could be easily visually detected even without molecular markers. You don't need to overemphasize your argument regarding use of molecular markers. Table 1. Based on the Lsat ID, it appears that originally at least 28 SSRs were used/tested. What happened to those not mentioned in the manuscript? Is “No tot = No F1 + No UG + No SP”. If yes, please indicate so. If not, please explain. Evanno’s calculation can’t determine if there is no population structure (K=1). Therefore, if K=2 is indicated as the best estimate, one may wonder if the result is accurate. Figures 3 and possibly also 4 do not add very much value and could be shown in supplemental files only. Alternatively, they could be combined into a single figure. L255, delete “in particular”. Fig 5a,b. Only nine bars are highlighted in red thought 10 lines are mentioned in the text. I assume that line 47 also should be highlighted, but because there is not variability in values, the bar is not shown. You need to identify/highlight this line in some other way. L279, replace “incredibly” with “only a”. L297, delete “interestingly”. L303, delete “extremely”. L306, replace “programming” with “planning”. L322, delete “It is of note that”. Journal name is missing. Those authors published more relevant paper than this in: DOI 10.1186/1471-2229-9-135 Figures S1 & S3. I would recommend deleting “%” sign from both tables. It will make them simpler and less cluttered. Then it can be mentioned in the legend that those values are in %.

Reviewer 3 Report

This manuscript describes in a very nice way a combination of classical and modern techniques. It will certainly be of interest to large scientific community as well those who follow classical breeding concepts as well as to those who promote modern ways of speeding breeding. Moreover, the publication can be used by companies and methods presented could be used in their breeding programs. This work can be also inspiring for other groups working with similar crops as well as groups having similar objectives. The presented methodology is straightforward, doesn't require fancy setup, as such can be also transferred to breeding programs in other parts of the globe.

Round 2

Reviewer 2 Report

The authors improved this manuscript by clarifying several points that were raised in the first review. A few remaining points are:

Major:

1. Authors provided more information regarding lettuce types, what is useful in understanding results. But not providing cultivars names (when available) due to confidentiality seems to be overly restrictive. All major journals require full disclosure of data, sequences, and all related information (including cultivar names). Because I am not familiar with the requirements of Genes, I will leave it up to the journal editor to decide if such information is mandatory.

2. P10, L347-348. I can’t agree that results from STRUCTURE and PCoA analyses show that “Parental germplasm appeared to derive from two different ancestors” as these analytical methods can identify population structure, but not the number of ancestors. Population structure and the number of ancestors is not the same.

Minor:

P10, L349-350. I guess it is more likely that your grouping didn't match type classification due to a very small number of molecular markers that were used for analyses. These methods (particularly STRUCTURE) are based on analyses of several thousand or at least several hundred of markers. You applied only 14 polymorphic markers that is very few to get reliable information. Supplemental Table S1. You may consider adding a column showing subpopulation classification for each of these lines.

Note (no correction is needed):

Your calculation of Evanno’s K=2 is correct, no argument here. However, since Evanno’s value is not defined for K=1, it is not possible to determine using this approach if there is no population structure (only a single, non-divided population). I raised this question because PCoA shows a possibility of no population structure and Evanno’s approach can’t disprove that notion.
